# Whey Cheeses Containing Probiotic and Bioprotective Cultures Produced with Ultrafiltrated Cow’s Whey

**DOI:** 10.3390/foods13081214

**Published:** 2024-04-16

**Authors:** Arona Pires, Agata Bożek, Hubert Pietruszka, Katarzyna Szkolnicka, David Gomes, Olga Díaz, Angel Cobos, Carlos Pereira

**Affiliations:** 1School of Agriculture, Polytechnic University of Coimbra, Bencanta, 3045-601 Coimbra, Portugal; arona@esac.pt (A.P.); david@esac.pt (D.G.); 2Departamento de Química Analítica, Nutrición y Bromatología, Facultad de Ciencias, Campus Terra, Universidade de Santiago de Compostela, 27002 Lugo, Spain; olga.diaz.rubio@usc.es (O.D.); angel.cobos@usc.es (A.C.); 3Department of Toxicology, Dairy Technology and Food Storage, West Pomeranian University of Technology, Al. Piastów 17, 70-310 Szczecin, Polandkatarzyna.szkolnicka@zut.edu.pl (K.S.); 4Centro de Estudos dos Recursos Naturais, Ambiente e Sociedade—CERNAS, 3045-601 Coimbra, Portugal

**Keywords:** whey cheese, ultrafiltration, probiotics, protective cultures

## Abstract

Bovine whey cheese (WC) is a product from southern European countries that presents some challenges: its production process involves high energy inputs; the yield is low; and WC has a short shelf life. The application of ultrafiltration (UF) to bovine whey before manufacture of WC and the employment of protective cultures can reduce these disadvantages. The objective of this research was the production of whey cheeses using ultrafiltrated bovine cheese whey with added probiotics or probiotics plus protective cultures. Three types of WC were produced: control CW without any addition (C); CW with the addition of the probiotic *Lactobacillus acidophilus* (LA5); and CW with the addition of *Lactobacillus acidophilus* plus a protective culture containing *Lacticaseibacillus paracasei* and *Lacticaseibacillus rhamnosus* (LA5FQ4). The WCs were stored under refrigerated conditions for 28 days. The products with added cultures presented lower pH values and higher titratable acidities when compared to the control. Sample LA5 presented the lowest pH and the highest titratable acidity, while LA5FQ4 presented intermediate values. Slight differences were observed between products regarding color parameters, chiefly resulting from storage time. The samples with added cultures were firmer when compared to the control, with LA5 cheeses showing the highest values at the end of the storage. Lactic acid bacteria (LAB) counts were on the order of log 8–9 CFU/g for the products with added cultures. Lower levels of yeasts and molds were detected on the sample with the protective culture (LA5FQ4), so that by the end of storage they presented counts one log cycle lower than C and LA5. Hence, the beneficial impact of the protective culture on the shelf life of the product is evident. Regarding sensory evaluation, LA5FQ4 cheeses obtained the highest scores for all parameters evaluated. It can be concluded that the use of UF associated with the addition of protective cultures can be very useful to reduce the energy consumption of the manufacturing process, to prolong the shelf life of WC and to improve its sensory properties.

## 1. Introduction

Cheese whey (CW) results from cheese production, and large amounts are generated worldwide. CW contains over 50% of the total solids found in the original whole milk; these include lactose, whey proteins, minerals and B vitamins [1,2]. About half of the CW generated worldwide is valorized through the production of whey powder, whey protein concentrates, whey protein isolates, whey protein hydrolysates or dried lactose [3,4]. In some countries, part of the whey is used to produce whey cheeses, known as *Requeijão* (Portugal), *Requesón* (Spain) or *Ricotta* (Italy) [5]. Whey cheese is a traditional product manufactured by heat treatment of whey, which allows for the recovery of most of the protein and fat fractions of CW [6]. Whey, previously acidified or not, is heated at 90–95 °C to precipitate whey proteins. The protein aggregates form a matrix that traps the fat and other solids that are present in the whey. The liquid remaining after production of whey cheeses is called “deproteinized” whey [7]. Whey cheeses are mostly produced with whey resulting from the production of sheep’s or goat’s milk cheeses. However, cow’s whey can also be used to produce such cheeses, although the yield is lower. Despite the interesting nutritional value of whey cheeses, only a small fraction of the CW produced is valorized as whey cheese. Only about 15% of the CW obtained annually in Italy is used in the production of *Ricotta* cheese [8]. The low valorization of whey through the production of whey cheese (WC) results from the fact that WC production is an energy-intensive process, which is associated with the short shelf life of the product and represents a challenge for producers.

To overcome the difficulties related to the low yield and high energy consumption associated with WC production, selective concentration of CW by ultrafiltration (UF) can represent a major aid for producers. Tangential filtration technologies such as UF or nanofiltration (NF) can be considered the best available tools to manage dairy byproducts. These technologies allow the selective concentration of specific components of whey, namely proteins. The proteins and fat of the CW are concentrated in the retentate, while water, lactose and salts permeate through the membrane; the main solid component of the permeate is lactose. The concentration of lactose and salts of both the retentate and permeate remain similar to that of the original CW [4]. So, this selective concentration process offers advantages over evaporation for the production of fermented dairy products, since high levels of lactose can originate products with high acidity. CW is concentrated by UF prior to whey cheese manufacturing, drastically reducing the volume of product to be submitted to the thermal treatment applied to aggregate whey proteins, thus reducing the energy required to produce whey cheeses [9,10,11,12,13,14,15]. Some authors also propose the use of transglutaminase to increase the yield in whey cheese production [16].

Several attempts have also been made to extend the shelf life of whey cheeses, most of them based on the use of modified atmosphere packaging [17,18,19,20] and, more recently, using lactic acid bacteria, including probiotics, as bioprotective cultures [21,22,23,24,25,26]. Regarding the incorporation of probiotic strains in *Requeijão*, considerable research has been performed by Madureira and coworkers [27,28,29,30,31,32,33,34]. However, color, textural and sensory properties of the resulting probiotic whey cheeses can be different than those of control WC, which can lead to lower acceptance by consumers. In a previous work, we evaluated the characteristics of cow’s WC containing probiotics that was produced with whey concentrated by UF [11]. To the best of our knowledge, this is the only work in which such a strategy was applied. In that research, we used kefir and bifidobacterium cultures with the aim of increasing the shelf life of WC. However, it was observed that the counts of yeasts and molds quickly increased after seven days of refrigerated storage. So, it was concluded that other techniques, such as modified atmosphere packaging (MAP), should be applied to increase the shelf life of whey cheeses. In the present work, we tested an innovative approach, namely the use of probiotics together with protective cultures. This solution may also represent an interesting possibility to increasethe shelf life of WC at lower costs, when compared to MAP.

Hence, the objective of this work was to produce whey cheeses with added probiotics or probiotics plus protective cultures, using UF concentrated bovine cheese whey. The products were compared with control whey cheeses without added cultures.

## 2. Materials and Methods

### 2.1. Production of Liquid Whey Concentrates

Sweet cheese whey resulted from the production of cow’s milk cheese at the dairy pilot plant of the Polytechnic Institute of Coimbra’s School of Agriculture (Coimbra, Portugal). CW (200 L) was submitted for ultrafiltration (UF) in a pilot plant (Proquiga Biotech SA, A Coruña, Spain) using a 3838 polyvinylidenefluoride/polysulfone UF membrane (effective filtration area of 7 m^2^; 10 kDa cut-off) supplied by FipoBiotech (Vigo, Spain). The membrane cut-off was selected because it retains almost all of the fat and protein of cheese whey while allowing for high filtration fluxes. The process was conducted at 40–45 °C with a transmembrane pressure of 3.0–3.5 bar, aiming to achieve a volumetric concentration factor (VCF = vol. feed/vol. retentate) of 5. After the UF concentration step, 40 L of liquid whey concentrate (LWC) was obtained.

### 2.2. Manufacture of Whey Cheeses

A total of 2% (*v*/*v*) full-fat milk (3.2% protein; 3.6% fat), 1% (*w*/*v*) salt, 2% (*w*/*v*) starch (Pregeflo™ CH 20, Roquette Freres, Lestrem, France) and 0.04% (*v*/*v*) CaCl_2_ (51% *w*/*v*, supplied by Tecnilac, Viseu, Portugal) was added to the LWC. The mixture was heated to 90–95 °C for 20 min to allow coagulation and subsequent precipitation of whey proteins. The precipitated fraction was separated from the remaining liquid and divided into three similar portions which were submitted to different treatments: control CW without any addition (C); CW with the addition of 10% (*w*/*w*) full-fat milk fermented with the probiotic *Lactobacillus acidophilus* (LA5™ Chr. Hansen, Hoersholm, Denmark) (LA5); and CW with the addition of 10% (*w*/*w*) full-fat milk fermented with the probiotic culture (LA5) plus a protective culture containing *Lacticaseibacillus paracasei* and *Lacticaseibacillus rhamnosus* (FreshQ4™, Chr. Hansen, Hoersholm, Denmark) (LA5FQ4). Both cultures were previously inoculated in the milk and incubated for 15 h at 30 °C according to the manufacturer’s instructions. 

The probiotic cultures were selected based on their commercial availability and proven activity. According to the manufacturer, the bioprotective culture has a combination of traditional lactic acid bacteria that inhibits unwanted yeasts and molds in fermented dairy products, and its effect is obtained through active participation in the natural fermentation process.

The whey cheeses were packaged in closed polypropylene boxes (Ondipack™, Makro, Carnaxide, Portugal) and stored under refrigerated conditions at 4 ± 1 °C for 28 days. Consecutive analyses were performed on the 1st, 7th, 14th, 21st and 28th days of storage. 

### 2.3. Physicochemical Analysis

Dry matter of samples was determined by drying according to NP 703:1982 for yogurt [35] in an oven (Schutzart DIN 40050-IP20 Memmert™, Schwabach, Germany). Ashes were determined by incineration of dry samples in an HD-23 Hobersal™ electric muffle furnace. The fat content was determined using the Gerber method (SuperVario-N Funke Gerber™ centrifuge, Berlin, Germany) in accordance with NP 2105:1983 [36]. Total nitrogen content was quantified by the Kjeldahl method (Digestion System 6 1007 Digester Tecator™, Foss Analytical, Häganäs, Sweden) following the AOAC (1997) standard [37]. The percentage of protein was calculated using a conversion factor of 6.38. The analyses were conducted in triplicate.

#### 2.3.1. pH and Titratable Acidity

The pH values were directly determined with the aid of an HI 9025 HANNA Instruments pH meter immediately after production and on the 1st, 7th, 14th, 21st and 28th days of storage. Acidity was determined according to NP 701:1982 for yogurts [38] by titration with a 0.1 N NaOH solution and was expressed as the percentage of lactic acid.

#### 2.3.2. Color Parameters

The color of whey cheese samples was determined with a colorimeter (Minolta™ Chroma Meter, model CR-200B, Tokyo, Japan) calibrated with a white standard (CR-A47: Y = 94.7; x 0.313; y 0.3204) using illuminant C and a 1 cm diameter aperture and a 10° standard observer. Color coordinates were measured in triplicate in the CIEL*a*b* system. Color difference (ΔEab*) was calculated as the following:ΔEab* = [(L* − L*^0^)^2^ + (a* − a*^0^)^2^ + (b* − b*^0^)^2^]^1/2^
(1)
where L*^0^, a*^0^ and b*^0^ and L*, a* and b* were the values measured for the samples under comparison. 

#### 2.3.3. Rheological Parameters

Rheological tests were performed at 5 °C with the aid of a Rheostress 6000 rheometer equipped with a plate-to-plate TMP35 bottom plate and a P35 TiL upper plate. Data acquisition was carried out using the HAAKE RheoWin Job Manager software (version 4.82.0002, ThermoHaake™, ThermoFisher Scientific, Waltham, MA, USA). The measuring system utilized a cone and plate geometry, specifically C60/Ti-0.052 mm (35 mm diameter and 1° angle). Stress sweep tests were conducted at 1 Hz to determine the linear viscoelastic rheological behaviors of all samples. The measurements were made on the 1st, 14th and 28th days of storage. Values for the elastic and viscous moduli (G′ and G″) and the complex viscosity (η*) were obtained at frequencies ranging from 0.05 to 1.0 Hz. 

The samples’ complex viscosities were fitted to the power law model according to Equation (2), and the consistency index *k* (Pa.s*^n^*^−1^) and the power law index (*n*) were determined. This model is typically used for shear-thinning fluids such as weak gels and low-viscosity dispersions.
η* = *k* (*ṙ*) *^n^*^−1^(2)

*k* = flow consistency index *ṙ* = shear strain *n* = flow behavior index (Pa.s^*n*^)

#### 2.3.4. Texture Parameters 

The textural properties of WC samples were evaluated using a Stable Micro Systems™ texture analyzer (mod. TA.XT Express Enhanced, Godalming, UK) using an acrylic cylindrical probe (25.4 mm diameter, 38.1 mm height) with a penetration distance of 15 mm at 2 mm/s. The results were calculated using the Specific Expression PC software (version 1.1.9.0). 

### 2.4. Microbiological Analysis

The microbial counts of lactic acid bacteria (LAB) were evaluated at the 1st, 7th, 14th, 21st and 28th days of storage. Microbial enumeration was performed on plates incubated at 37 °C for 48 h on M17 agar for presumptive lactococci (in aerobic conditions) and on MRS agar (under anaerobic conditions) for presumptive lactobacilli (including lacticaseibacilli) according to ISO 7889, IDF 117 (2003) [39]. Yeasts and molds were counted in accordance with ISO 6611, IDF 94 (2204) [40] on plates incubated at 25 °C. Culture media were supplied by Biokar Diagnostics (Allonee, France). The results are expressed as log CFU/g of product, and the analyses were performed in triplicate with two controls for each medium. 

### 2.5. Sensory Analysis

Analysis of the sensory characteristics was conducted in isolated booths with the same lighting and temperature conditions. In the acceptance test, the samples, identified by three-digit random numbers, were rated for aroma, taste, texture and overall impression using a 9-point hedonic scale, where 1 = dislike extremely and 9 = like extremely. The evaluation was carried out using a non-trained panel consisting of 30 members [41]. The panel members were also asked to rank the samples in accordance with their preference, from most preferred (1) to least preferred (3), according to ISO 8587 (1988) [42]. The members of the panel were informed of the objectives of the test and signed the informed consent form in use at the Polytechnic of Coimbra, giving their consent for the treatment and publication of results.

### 2.6. Statistical Analysis

The influence of storage time, the kind of whey cheese and their interaction on all the determined parameters were analyzed by a two-way ANOVA. The differences among whey cheeses and among different days of storage were assessed by one-way ANOVA and the means were compared using the Tukey’s post hoc test. For all mean evaluations, a significance level of *p* < 0.05 was used (IBM SPSS Statistics for Windows, version 27, 2021; IBM Corp, Armonk, NY, USA). Graphs were generated using the Statistica Software, version 8 (Stasoft Europe, Hamburg, Germany). 

## 3. Results

Table 1 represents the protein, fat and ashes contents of the different WC samples. Non-significant differences were observed regarding the protein content of whey cheeses, but the products with added microbial cultures presented higher fat and lower mineral contents. These differences probably resulted from the added milk in which the cultures were pre-incubated (which was added at a level of 10% *w*/*w* of the whey cheeses). 

The changes in the dry matter and in the water activity of the products are presented in Figure 1. Significant differences were observed regarding the evolution of the dry matter of samples, either between products or for the same product at different days of storage. The samples LA5 and LA5FQ4 presented slightly higher solids contents when compared to the control, except at the 14th day of storage. However, the two-factor ANOVA did not indicate differences resulting from the interaction of factors. Differences were also observed regarding the water activity of whey cheese samples, chiefly during the first 7 days of refrigerated storage (Appendix A). 

Significant differences were also observed regarding the pH values and titratable acidities (Figure 2 and Table 2 and Appendix A). As expected, the products with added cultures presented lower pH values and higher titratable acidities when compared to the control. The sample LA5 presented the lowest pH and the highest titratable acidity. LA5FQ4 presented intermediate values (pH 5.47 and TA 0.33% lactic acid at the end of storage). These values probably played an important role regarding the better acceptance of this sample by consumers, as will be discussed later.

The evolution of the color parameters of the whey cheeses is shown in Figure 3. The lightness (L*) ranged between 80 and 90 over the storage duration. The lowest L* values were observed for sample LA5FQ4 (Appendix A). The a* values decreased between the 1st and the 14th day, then tended to increase slightly. Conversely, the b* values increased from the 1st to the 14th day, then decreased.

A better indicator of the differences in color are the values of ΔEab* (Table 3). It can be observed that no color differences were detected between the control and LA5 at the 1st, 21st and 28th days (values < 1), while the sample LA5FQ4 was like the control at the 21st day of storage, being different at all other days at which they were compared. Significant differences were observed for the same sample at different days of storage, with this trend being more evident between the 7th and the 1st day, chiefly for the control (C) and for LA5FQ4. The sample LA5 presented lower differences between the 1st and the 7th day of storage and between the 14th and the 7th day, as well as between the 21st and the 14th day. In Table 4, it can be observed that the L* parameter was not different between products (*p* = 0.176) nor between storage times (*p* = 0.101). Additionally, no differences were observed between products regarding parameters a* (*p* = 0.815) and b* (*p* = 0.087).

Figure 4 presents the rheological parameters of whey cheese samples at the 1st, 14th and 28th days of storage. The elastic modulus (G′) was always higher than the viscous modulus (G″). Hence, the loss tangent, or tan δ (ratio of G″/G′), was lower than 1 (C—0.187 ± 0.017; LA5—0.230 ± 0.024; LA5FQ4—0.211 ± 0.013). These values show that the samples had mainly elastic behavior. Whey cheese samples exhibited a flow behavior that was dependent on shear as the complex dynamic viscosity (η*) decreased linearly with the increase in frequency on a double logarithmic scale (Figure 4B,D,F). So, whey cheese samples showed a non-Newtonian shear-thinning or pseudoplastic behavior. Additionally, the power law parameters indicated the pseudoplastic, or shear-thinning, nature of the products (*n* < 1). Pseudoplastic fluids are those whose behavior is independent of time and that have a lower apparent viscosity at higher shear rates. In general, these fluids are solutions of large, polymeric molecules in a solvent with smaller molecules. An explanation for such behavior is that the large molecular chains tumble randomly, affecting large volumes of fluid at low shear, but gradually align in the direction of increasing shear and produce less resistance, and consequently lower dynamic viscosity values. Table 5 presents the regression equations for the log/log plots of dynamic viscosity/frequency. Sample LA5 presented significantly higher values for G′, G″ and η* over the storage period. This is evident in Table 6, which displays the calculated power law parameters.

The firmness of the samples was also determined using a texturometer (Figure 5). The determined values agreed with the rheological parameters obtained with the rheometer. The samples with added cultures were firmer when compared to the control throughout the storage period. These differences were more marked at the beginning and at the end of the storage period. Sample LA5 presented the highest values by the end of storage.

The microbiological characteristics of the whey cheeses are displayed in Figure 6 (and in Appendix A). Lactococci and lactobacilli (including lacticaseibacilli) counts were on the order of log 8–9 CFU/g for the products with added cultures. The control sample presented levels of lactococci and lactobacilli of ca. 6.5 and 5.5 log CFU/g, respectively, at the beginning of storage, but their counts increased steadily towards the end of refrigerated storage (to ca. log 8 CFU/g). It is worth mentioning the lower levels of yeasts and molds detected on the sample with the protective culture (LA5FQ4), which by the end of storage presented counts one log cycle lower than C and LA5. Hence, the beneficial impact of the protective culture (FQ4™) on the shelf life of the product is evident. The high levels of presumptive lactococci that were observed probably resulted from the low selectivity of the M17 culture media, which allows for the growth of other lactic acid bacteria, or even non-LAB, as reported by other authors [43,44]. Table 7 presents the two-way ANOVA test and clearly indicates the significant differences in the microbiota of whey cheeses among products and among different storage times.

Figure 7 presents the sensory evaluation of the whey cheeses. Significant differences were found regarding the taste (*p* = 0.0031) and the global evaluation (*p* = 0.0032). No differences were observed for the aroma (*p* = 0.1963) or for the texture (*p* = 0.4455) of the products. Whey cheese containing probiotics (LA5) obtained lower scores for the taste and for the global evaluation, while the sample LA5FQ4 obtained the highest scores for all evaluated parameters. The ranking test also differentiated this sample in comparison with the control (C) and LA5, which were ranked second and third, respectively. The higher acidity of sample LA5 was the reason for its lower acceptance.

## 4. Discussion

Whey cheeses showed lower values for dry matter, protein and fat than those observed in a previous work [11], in which whey cheeses with added fermented cream as the vehicle for probiotics presented a higher level of total solids (22–25% *w*/*w*), protein (33–36%, dry basis) and fat (30–33% dry basis). These differences probably resulted from the type of product used to pre-incubate the cultures (cream). Garcia and coworkers prepared goat’s probiotic whey cheeses using a similar approach and reported higher values of dry matter (25–27%), but lower average values for fat (25.91 ± 4.05%, dry basis) and protein (43.18 ± 1.96%, dry basis) contents [6]. 

The color values were different than those reported in whey cheeses with kefir or probiotics [11], where the values of L* were higher (between 94 and 96), a* values were lower (between −3.0 and −3.5) and b* values were higher (between 8 and 9). We have not found studies about the effect of inclusion of probiotic bacteria in the color of WC.

Regarding the rheological properties, the whey cheeses with added cultures were firmer when compared to the control products. On the contrary, Pires et al. [11] observed that the addition of probiotics and kefir caused decreases in the elastic modulus (G′) of the whey cheeses. They indicated that the cultures used (*Bifidobacterium* spp. and a commercial kefir culture) could weaken the protein structure. In the present study, the cultures of *Lactobacillus acidophilus*, *Lacticaseibacillus paracasei* and *Lacticaseibacillus rhamnosus* did not affect the protein structure. On the contrary, the whey cheeses with *Lactobacillus acidophilus* showed higher firmness. Madureira et al. [30,32] studied the rheological characteristic of whey cheeses manufactured with probiotic bacteria *Lacticaseibacillus casei* [30] or *Lacticaseibacillus casei* and *Bifidobacterium animalis* [32]. They reported that in all the whey cheeses, the elastic modulus (G′) was higher than the viscous modulus (G″), typical of solid viscoelastic foods. They also observed, as in our study, that the probiotic whey cheeses were firmer, more elastic and more viscous than non-inoculated whey cheeses. They indicated that this was due to the lower pH of these cheeses, indicating that there is a positive correlation between acidity and hardness of cheeses and that higher acidity values lead to firmer matrices. In the cheeses of our study, the whey cheeses with added cultures had higher titratable acidities and lower pH values than control whey cheeses, and consequently presented higher firmness; LA5 whey cheeses showed the highest firmness values by the end of refrigerated storage and had the lowest pH.

Madureira et al. [30,32] also indicated that higher acidity values negatively affected the sensory acceptance of probiotic whey cheeses. In the present study, despite LA5FQ4 whey cheeses having higher acidities than the control whey cheeses, they obtained higher sensorial evaluations. It was also reported that whey cheeses inoculated with cultures of *Lactobacillus* spp. have better sensory scores than whey cheeses with *Bifidobacterium* spp. due to the production of acetic acid by the latter, which has negative consequences in sensorial evaluation [32].

Results for viable cell counts of goat’s probiotic whey cheeses (containing *Lacticaseibacillus rhamnosus* and *Bifidobacterium animalis*) presented an average of log 8.24 CFU/g on the final day of storage [6], whereas in the present study microbial counts of lactic acid bacteria (LAB) were on the order of log 9 CFU/g for LA5FQ4 and log 8 CFU/g for LA5. Although the levels of probiotic bacteria that have beneficial consequences for health are normally strain- and host-dependent, it has been indicated that concentrations of at least log 7 CFU/g are necessary for a positive effect [45,46]. Hence, it can be concluded that even at the end of storage, probiotic counts of LA5 and of LA5FQ4 were at levels that allow for that claim. 

It must be noted that by the end of storage, the control sample presented values higher than log 8 CFU/g for bacteria counted on M17 and MRS agar, but these values resulted from adventitious contamination and subsequent growth during storage. However, during the first two weeks of storage the counts were much lower than the ones observed in the samples with added cultures, and therefore the acidity values were lower. Another possibility for the lower acidity of the control sample can be related to a lower capacity for production of lactic acid by the bacteria present in this sample. Whey cheeses with kefir culture or *Bifidobacterium* spp. showed lower counts of lactic acid bacteria (log 7 UFC/g) after 21 days of storage [11]. These results may indicate that the cultures used in the present study have a better adaptation to the whey cheese matrix. Additionally, Pires et al. [11] did not observe reduction in the counts of yeasts and molds in the whey cheeses with *Bifidobacterium* spp. and commercial kefir culture in relation to control whey cheeses, while in the present study the protective culture reduced yeast and mold counts by one log cycle. The protective culture with *Lacticaseibacillus paracasei* and *Lacticaseibacillus rhamnosus* (together with *Lactobacillus acidophilus)* showed a better effect on the shelf life of the product than the culture with only the probiotic *Lactobacillus acidophilus*.

The research of Madureira and coworkers indicated that whey cheese protected the probiotic strains during transit throughout the simulated gastrointestinal system, therefore comprising a promising carrier of those bacteria. Probiotic bacteria (*Lacticaseibacillus casei* and *Bifidobacterium animalis*) added to whey cheese also showed positive effects concerning control of spoilage and pathogenic microorganisms, thus promoting the extension of the shelf life of the product [28,31,33]. Hence, the approach of adding probiotic or probiotic plus protective cultures to whey cheeses can increase the shelf life of whey cheeses while providing beneficial health effects to consumers.

## 5. Conclusions

The concentration of cow’s whey by ultrafiltration proved to be a very useful tool to reduce the volume of whey to be submitted to the thermal treatment applied to precipitate the whey proteins. Using a volumetric concentration factor of 5, one can estimate that the energy requirements for the process can be reduced by over 70%. The probiotic whey cheeses (with *Lactobacillus acidophilus* and with a protective culture containing *Lacticaseibacillus paracasei* and *Lacticaseibacillus rhamnosus*) manufactured using UF-concentrated bovine cheese whey showed the lowest levels of yeasts and molds at the end of the refrigerated storage. However, the probiotic whey cheeses showed some differences (lower pH values, higher titratable acidities and firmer textures) when compared to control whey cheeses. The product with a probiotic plus a bioprotective culture obtained the best global sensory evaluation. These results indicate that the addition of protective cultures with *Lacticaseibacillus* strains can be very useful to prolong the shelf life of whey cheeses while also improving the sensory evaluation of whey cheeses and providing the health benefits associated with the consumption of probiotics.

## Figures and Tables

**Figure 1 foods-13-01214-f001:**
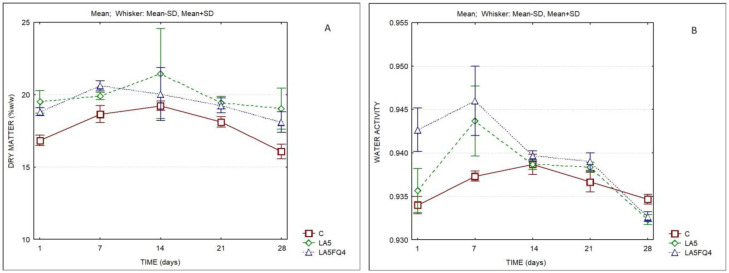
Dry matter and water activity of whey cheeses. C—control; LA5—whey cheese with probiotic LA5 culture; LA5FQ4—whey cheese with probiotic LA5 culture plus FQ4 protective culture. (**A**) Dry matter; (**B**) Water activity.

**Figure 2 foods-13-01214-f002:**
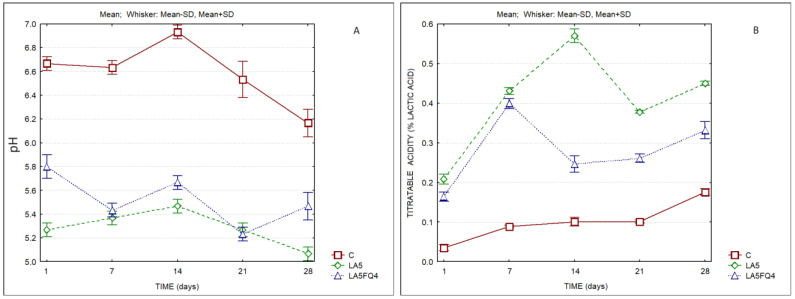
pH and titratable acidity of whey cheeses. C—control; LA5—whey cheese with probiotic LA5 culture; LA5FQ4—whey cheese with probiotic LA5 culture plus FQ4 protective culture. (**A**) pH; (**B**) Titratable acidity.

**Figure 3 foods-13-01214-f003:**
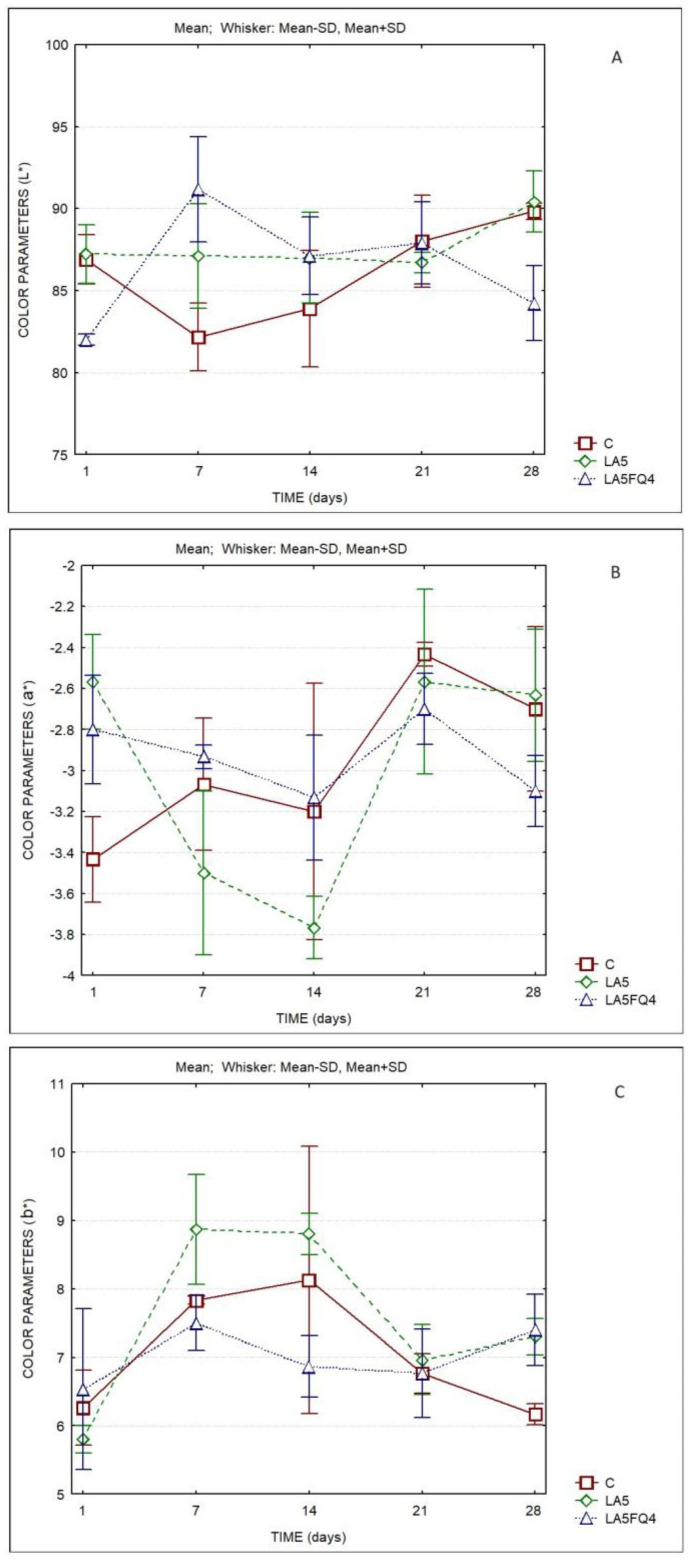
Color parameters (CIEL*a*b*) of whey cheeses. C—control; LA5—whey cheese with probiotic LA5 culture; LA5FQ4—whey cheese with probiotic LA5 culture plus FQ4 protective culture. (**A**) L*; (**B**) a*; (**C**) b*.

**Figure 4 foods-13-01214-f004:**
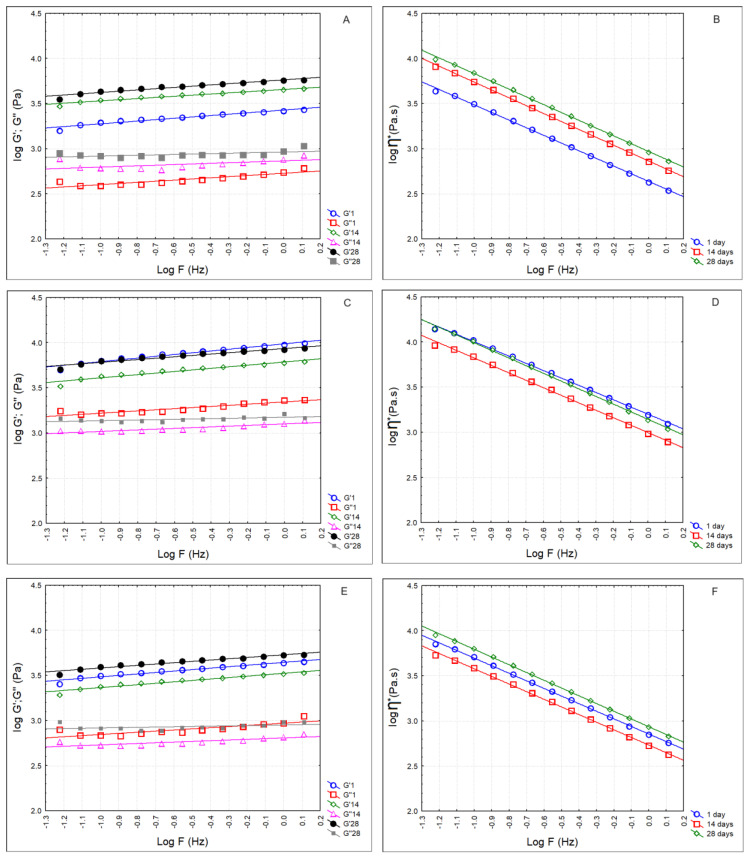
Rheological parameters of whey cheeses over storage. (**A**) G′—elastic modulus; G″—viscous modulus; (**B**) η*—dynamic viscosity of control cheese. (**C**) G’—elastic modulus; G″—viscous modulus; (**D**) η*—dynamic viscosity of LA5 (whey cheese with probiotic LA5 culture); (**E**) G′—elastic modulus; G″—viscous modulus; (**F**) η*—dynamic viscosity of LA5FQ4 (whey cheese with probiotic LA5 culture plus FQ4 protective culture).

**Figure 5 foods-13-01214-f005:**
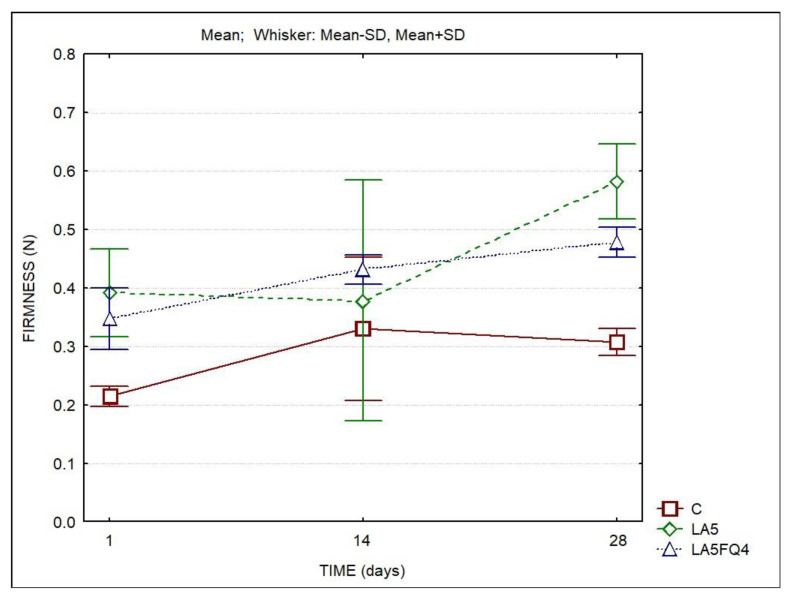
Firmness of whey cheeses over storage. C—control; LA5—whey cheese with probiotic LA5 culture; LA5FQ4—whey cheese with probiotic LA5 culture plus FQ4 protective culture.

**Figure 6 foods-13-01214-f006:**
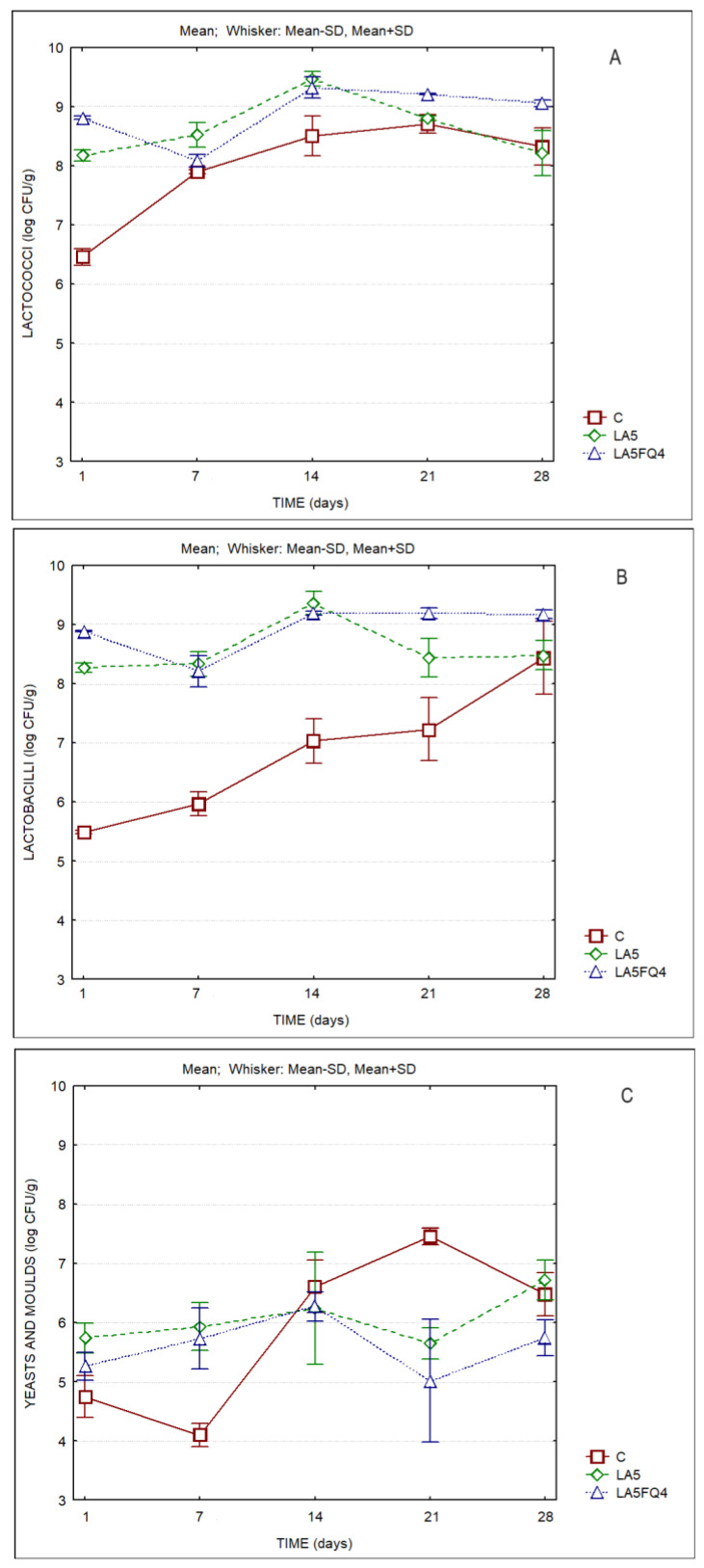
Microbial parameters of whey cheeses over the storage period. (**A**) Lactococci; (**B**) lactobacilli; (**C**) yeasts and molds. C—control; LA5—whey cheese with probiotic LA5 culture; LA5FQ4—whey cheese with probiotic LA5 culture plus FQ4 protective culture.

**Figure 7 foods-13-01214-f007:**
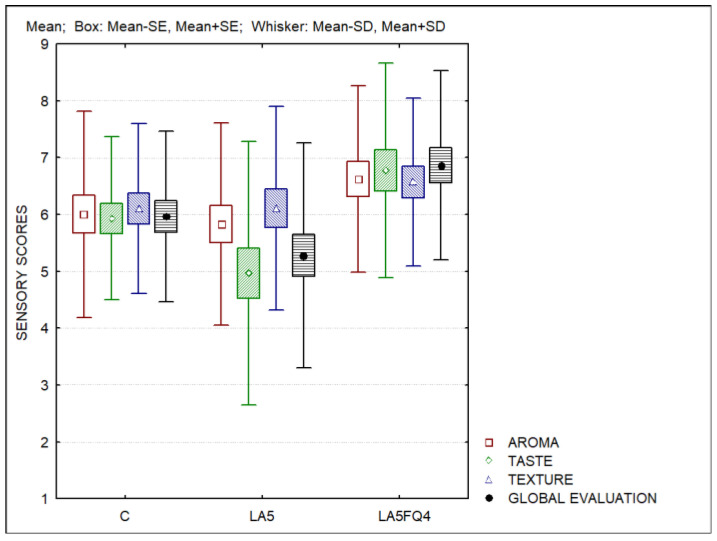
Sensory evaluation of whey cheeses. C—control; LA5—whey cheese with probiotic LA5 culture; LA5FQ4—whey cheese with probiotic LA5 culture plus FQ4 protective culture.

**Table 1 foods-13-01214-t001:** Gross chemical composition of whey cheeses. C—control; LA5—whey cheese with probiotic LA5 culture; LA5FQ4—whey cheese with probiotic LA5 culture plus FQ4 protective culture.

	Protein in Dry Matter (% *w*/*w*)	Fat in Dry Matter (% *w*/*w*)	Ashes in Dry Matter (% *w*/*w*)
C	29.75 ± 2.07 ^a^	21.07 ± 0.41 ^a^	7.72 ± 0.79 ^a^
LA5	33.08 ± 4.59 ^a^	24.41 ± 1.36 ^b^	6.18 ± 0.47 ^b^
LA5FQ4	33.94 ± 0.06 ^a^	24.24 ± 0.24 ^b^	6.71 ± 0.38 ^b^

^a,b^: Same superscript letter (a) in a column indicates no significant differences.

**Table 2 foods-13-01214-t002:** Two-way ANOVA of physicochemical parameters of whey cheese samples.

		Two-Way ANOVA	
		F	*p* Value
Dry matter	Time	7.70	0.000
	Product	15.08	0.000
	Interaction	0.67	0.715
a_w_	Time	27.85	0.000
	Product	15.31	0.000
	Interaction	5.83	0.000
pH	Time	45.81	0.000
	Product	1118.10	0.000
	Interaction	11.34	0.000
Titratable acidity	Time	368.49	0.000
	Product	2508.79	0.000
	Interaction	109.05	0.000

**Table 3 foods-13-01214-t003:** ΔEab* values between products and over time for each product.

ΔEab* Values between Products					
Storage Time (days)	1	7	14	21	28
Products					
LA5 vs. C	0.54	15.24	5.19	0.87	0.82
LA5FQ4 vs. C	12.24	47.75	6.03	0.04	16.52
**ΔEab* Values for the Same Product at Different Storage Days**					
		C	LA5	LA5FQ4	
7 d vs. 1 d		23.96	5.15	42.49	
14 d vs. 7 d		6.27	5.25	13.29	
21 d vs. 14 d		1.23	0.82	17.44	
28 d vs. 21 d		4.58	6.25	2.91	

**Table 4 foods-13-01214-t004:** Two-way ANOVA of color parameters of whey cheese samples.

CIEL*a*b* Color Parameters		Two-Way ANOVA	
		F	*p* Value
L*	Time	2.14	0.101
	Product	1.84	0.176
	Interaction	5.81	0.000
a*	Time	8.85	0.000
	Product	0.21	0.815
	Interaction	3.71	0.004
b*	Time	10.82	0.000
	Product	2.66	0.087
	Interaction	2.38	0.041

**Table 5 foods-13-01214-t005:** Regression equations for the log/log plots of dynamic viscosity/frequency (r^2^ > 0.99).

Products/Storage Time	C	LA5	LA5FQ4
1 day	y = 2.637 − 0.849 × x	y = 3.200 − 0.808 × x	y = 2.855 − 0.842 × x
14 days	y = 2.863 − 0.876 × x	y = 2.996 − 0.830 × x	y = 2.733 − 0.846 × x
28 days	y = 2.969 − 0.864 × x	y = 3.143 − 0.852 × x	y = 2.936 − 0.859 × x

C—control; LA5—whey cheese with probiotic LA5 culture; LA5FQ4—whey cheese with probiotic LA5 culture plus FQ4 protective culture.

**Table 6 foods-13-01214-t006:** Power law parameters of whey cheese samples.

Products	*k* (kPa.s)	*n*	r^2^
C	4.46 ± 1.45	0.972 ± 0.016	0.869
LA5	11.72 ± 4.45	0.977 ± 0.089	0.875
LA5FQ4	5.06 ± 0.92	0.989 ± 0.009	0.871

C—control; LA5—whey cheese with probiotic LA5 culture; LA5FQ4—whey cheese with probiotic LA5 culture plus FQ4 protective culture.

**Table 7 foods-13-01214-t007:** Two-way ANOVA of microbiological parameters of whey cheese samples.

Microbial Groups		Two-Way ANOVA	
		F	*p* Value
Lactococci	Time	68.77	0.000
	Product	48.47	0.000
	Interaction	18.54	0.000
Lactobacilli	Time	170.83	0.000
	Product	23.10	0.000
	Interaction	10.77	0.000
Yeasts and Molds	Time	4.58	0.014
	Product	14.13	0.000
	Interaction	8.31	0.000

## Data Availability

Data from physicochemical and color parameters and microbiological counts are available in the manuscript and Appendix A. Further inquiries can be directed to the corresponding author.

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
