# Peer review of "Whey Cheeses Containing Probiotic and Bioprotective Cultures Produced with Ultrafiltrated Cow’s Whey"

_foods, 2024, doi:10.3390/foods13081214_

Round 1

Reviewer 1 Report

Comments and Suggestions for Authors

Whey cheese has several beneficial effects on human health. But, production of whey cheese need high energy and the product yield is low.

Ultrafiltration technology has great potential to increase the yield, for instance (because of the selective concentration).

Protective cultures are suitable to prolong the shelf life of the product. Therefore, the topic of the manuscript can be considered as relevant not just for the science but also for the industry practice. The applied methods are adequate to sample characteristics and the specific aims of the research. Statistical methods are applied for the analysis, as well.

The manuscript contains significant and interesting results that are represented clearly and discussed in details with relevant references. The manuscript has high scientific quality in general, but need revision (see my comments).

Comments, suggestions:

Please discuss briefly the ’general’ efficiency of UF for whey fractionation/concentration int he Introduction section, as well.

Please define and give clearly the novelties of the study in the Introduction section.

Please give how the UF membrane selected for whey concentration (material, cut-off, etc.) was.

Please give how the probiotic culture selected was.

The visibility of Figure 1-3 is poor. Please improve it (mainly axis titles, scales).

The Conclusion section is too general. Please rephrase it to be more concrete.

Author Response

Dear reviewer

Thank you for your comments and suggestions that certainly allowed us to improve the quality of the manuscript. We hope that our response is satisfactory.

Reviewer 2 Report

Comments and Suggestions for Authors

1. According to the latest taxonomic status and Latin name, please change Lactobacillus paracasei and Lactobacillus rhamnosus in line 18 on the first page to  Lacticaseibacillus paracasei and  Lacticaseibacillus rhamnosus. Please check the full text for replacement.

2. In Figure 2, the LA5 product has the highest pH at 14 days, but the corresponding pH value is also the highest. Why are there such contradictory results? Normally, the lower the pH, the higher the acidity. In addition, why did the acidity of LA5 and LA5FQ4 cheeses in Figure 2B decrease at a later stage?

3. In Figure 6, why are the initial numbers of lactococci in LA5 and LA5FQ4 cheeses so much higher than those in the control group? The inoculations all use bacillus fermentation. What is the source of the lactococci?

4. As can be seen from Figure 6A and B, the numbers of Lactobacillus and Lactococcus in the control group are increasing rapidly. Both of them are lactic acid bacteria, which will inevitably produce acid, causing their pH to drop. However, in Figure 2, the pH change is It first increases and then decreases. When the pH increases, does it mean that there are fewer hydrogen ions? Why does this result occur?

5. It can be seen from Figure 6A and B that at the end of storage, the number of Lactobacillus and Lactococcus in the control group is consistent with the number of LA5. The growth of the bacteria must metabolize and produce lactic acid, but its acidity is only half of that of the LA5 group? Why?

Author Response

(The authors gave the same response as above.)
